# Determinants of In-Hospital Mortality in Elderly Patients Aged 80 Years or above with Acute Heart Failure: A Retrospective Cohort Study at a Single Rural Hospital

**DOI:** 10.3390/jcm10071468

**Published:** 2021-04-02

**Authors:** Yusuke Watanabe, Kazuko Tajiri, Hiroyuki Nagata, Masayuki Kojima

**Affiliations:** 1Department of Internal Medicine, Hitachiomiya Saiseikai Hospital, 3033-3 Tagouchichou, Hitachiomiya, Ibaraki 319-2601, Japan; m02067hn@jichi.ac.jp; 2Department of Cardiology, Faculty of Medicine, University of Tsukuba, Tsukuba 305-8577, Japan; ktajiri@md.tsukuba.ac.jp; 3Department of Surgery, Hitachiomiya Saiseikai Hospital, Hitachiomiya 319-2256, Japan; allforonekojima@gmail.com

**Keywords:** acute heart failure, elderly patients, SOB-ASAP score, phosphodiesterase 3 inhibitor, antibiotics

## Abstract

Heart failure is one of the leading causes of mortality worldwide. Several predictive risk scores and factors associated with in-hospital mortality have been reported for acute heart failure. However, only a few studies have examined the predictors in elderly patients. This study investigated determinants of in-hospital mortality in elderly patients with acute heart failure, aged 80 years or above, by evaluating the serum sodium, blood urea nitrogen, age and serum albumin, systolic blood pressure and natriuretic peptide levels (SOB-ASAP) score. We reviewed the medical records of 106 consecutive patients retrospectively and classified them into the survivor group (*n* = 83) and the non-survivor group (*n* = 23) based on the in-hospital mortality. Patient characteristics at admission and during hospitalization were compared between the two groups. Multivariate stepwise regression analysis was used to evaluate the in-hospital mortality. The SOB-ASAP score was significantly better in the survivor group than in the non-survivor group. Multivariate stepwise regression analysis revealed that a poor SOB-ASAP score, oral phosphodiesterase 3 inhibitor use, and requirement of early intravenous antibiotic administration were associated with in-hospital mortality in very elderly patients with acute heart failure. Severe clinical status might predict outcomes in very elderly patients.

## 1. Introduction

Heart failure is one of the most common diseases worldwide and generally affects the elderly [1,2,3]. Elderly patients with heart failure are likely to have many comorbidities [2]. The prevalence of heart failure is on the rise due to the rapidly aging population [3,4]; in 2019, the estimated number of individuals aged 65 years or above in Japan was approximately 36 million (28.4% of the total population). Heart failure is the leading cause of mortality in Japan. Although several novel medications have been developed [5], therapeutic strategies for improving patients’ prognoses are yet to be identified.

Previous studies have suggested that risk score systems are useful for predicting prognosis in inpatients and outpatients with heart failure [6,7,8]. Other studies have reported factors leading to in-hospital mortality, such as acute kidney injury, new-onset atrial fibrillation, and nutritional index [9,10,11]. However, most of these studies have focused on relatively younger populations than the Japanese elderly population, and very few studies have focused on the very elderly population [12,13]. The population of hospitalized patients with acute heart failure is aging, even in rural areas. Recently, a novel scoring system, the serum sodium, blood urea nitrogen, age and serum albumin, systolic blood pressure and natriuretic peptide level (SOB-ASAP) score, was developed in Japanese registries [14]. The SOB-ASAP score ranges from 0 to 14; the highest score indicates a high in-hospital mortality rate [14].

This study aimed to reveal other determinants of in-hospital mortality in very elderly patients (aged 80 years or older) with acute heart failure by evaluating their SOB-ASAP score.

## 2. Materials and Methods

This single-hospital retrospective observational study was conducted at the Hitachiomiya Saiseikai Hospital, Japan. We reviewed the medical records of all consecutive patients with heart failure admitted to our hospital between January 2017 and December 2019. The inclusion criteria were as follows: (1) hospitalized patients aged ≥80 years and diagnosed with heart failure, (2) clinical status corresponding to heart failure, according to the Framingham criteria [15], and (3) left ventricular function assessed with echocardiography, at least. The exclusion criteria were as follows: (1) brain natriuretic peptide (BNP) level <100 pg/mL or unknown, (2) readmission of the same patient with acute heart failure, (3) no diagnosis of acute heart failure, and (4) requiring transfer to a tertiary hospital. We divided the participants into two groups based on the prevalence of in-hospital mortality: the survivor group and the non-survivor group.

This study was approved by our institutional review board (ID 20-06) and was conducted in accordance with the Declaration of Helsinki for experiments involving humans. The requirement for written informed consent was waived by our institutional review board due to the retrospective nature of the study.

In echocardiography, data acquisition was performed by an expert. The variables measured and derived using echocardiography were determined as follows: two-dimensional left ventricular ejection fraction (LVEF) was computed from the calculated left ventricular end-diastolic and end-systolic volumes. Valvular heart disease was defined as moderate to severe valvular disease (according to current guidelines [16,17,18]) or a history of valvular surgery or cardiac surgery. Wall motion abnormality was defined as localized abnormal wall motion, such as akinesis, hypokinesis, and dyskinesis.

Hypertension was defined as the use of medication for hypertension and/or a history of hypertension before admission. Dyslipidemia was defined as a triglyceride level ≥150 mg/dL, low-density lipoprotein cholesterol level ≥140 mg/dL, high-density lipoprotein cholesterol level ≤40 mg/dL, the use of medication for dyslipidemia, or a history of dyslipidemia. Diabetes mellitus was defined as a hemoglobin A1c level ≥6.5% (National Glycohemoglobin Standardization Program value), the use of medication for diabetes mellitus, or a history of diabetes mellitus. Chronic obstructive pulmonary disease (COPD) was defined as the use of medical treatment for COPD and/or a history of COPD before admission. We calculated the estimated glomerular filtration rate (eGFR) from the serum creatinine levels, age, weight, and sex using the following formula: eGFR (mL/min/1.73 m^2^) = 194 × s-Cr (−1.094) × age (−0.287) × 0.739 (if female) [19]. Worsening renal function was defined as an increase in the serum creatinine level to >0.3 mg/dL [20]. The SOB-ASAP score was calculated according to the previously published formula [14].

### Statistical Analyses

All statistical analyses were performed using SPSS 26.0 for Windows (SPSS, Chicago, IL, USA).

Continuous data are expressed as mean ± standard deviation (SD). Normality was tested using the Shapiro–Wilk test. Normally distributed continuous variables were compared between the two groups using the unpaired Student’s *t*-test. Continuous variables were compared using the Mann–Whitney *U*-test. Categorical variables were expressed as numbers and percentages and were compared using the Pearson’s χ^2^ test or the Fisher’s exact test. Multivariate stepwise regression analysis was used to evaluate the in-hospital mortality in elderly patients (aged ≥ 80 years) with acute heart failure; the included variables were found to be significant (*p* < 0.1) using a univariate logistic regression analysis. We analyzed the relationship between LVEF and the in-hospital mortality by constructing a receiver operating characteristics (ROC) curve and calculating the area under the curve. The area under the curve was 0.66 (95% confidence interval (CI): 0.53–0.79, *p* = 0.018). The sensitivity and specificity for an LVEF of 49.6% were 60.2% and 39.1%, respectively. Therefore, an LVEF ≥ 50% was analyzed in the univariate analysis. Meanwhile, the variables included in the SOB-ASAP scoring system, such as the systolic blood pressure and BNP, were excluded from the multivariate model. A *p*-value < 0.05 was considered statistically significant. 

## 3. Results

A total of 106 patients (36.8% men, mean age: 89.8 ± 4.5 years) were included in the study (survivor group: *n* = 83; non-survivor group: *n* = 23) (Figure 1).

Table 1 shows the comparison of the baseline characteristics and medication usage at admission between the two groups. The systolic blood pressure was significantly higher in the survivor group than in the non-survivor group (139 ± 33 mmHg vs. 119 ± 20 mmHg, *p* = 0.011). A systolic blood pressure of ≤100 mmHg, indicating an unstable hemodynamic status, was more prevalent in the non-survivor group than in the survivor group (26.1% vs. 8.4%, *p* = 0.022). The SOB-ASAP score was significantly better in the survivor group than in the non-survivor group (4.3 ± 2.3 vs. 6.8 ± 2.7, *p* < 0.001). Mineralocorticoid receptor antagonist usage was significantly lower in the survivor group than in the non-survivor group (10.8% vs. 30.4%, *p* = 0.028).

Table 2 shows the differences in the baseline laboratory data and echocardiographic parameters between the two groups. The serum sodium levels were significantly higher in the survivor group than in the non-survivor group (139 ± 5 mEq/L vs. 136 ± 5 mEq/L, *p* = 0.025). The C-reactive protein level was significantly lower in the survivor group than in the non-survivor group (1.5 ± 2.3 mg/dL vs. 5.1 ± 7.2 mg/dL, *p* = 0.0073). The BNP level was significantly lower in the survivor group than in the non-survivor group (646.9 ± 586.9 pg/mL vs. 1170.7 ± 1018.8 pg/mL, *p* = 0.0033). The LVEF value was significantly better in the survivor group than in the non-survivor group (52.8 ± 17.5% vs. 42.1 ± 19.9%, *p* = 0.018).

Table 3 shows the comparison of treatment strategies and subsequent outcomes during hospitalization between the two groups. The prevalence of intravenous administration of diuretics within 48 h of hospitalization was significantly higher in the survivor group than in the non-survivor group (80.7% vs. 56.5%, *p* = 0.017). The prevalence of intravenous catecholamine support requirement and intravenous administration of antibiotics within 48 h of hospitalization was significantly lower in the survivor group than in the non-survivor group (1.2% vs. 13.0%, *p* = 0.031; 14.5% vs. 34.8%, *p* = 0.033, respectively). Intravenous catecholamine support requirement, intravenous administration of antibiotics, and noninvasive positive pressure ventilation support during the entire period of hospitalization were more prevalent in the non-survivor group than in the survivor group (3.4% vs. 30.4%, *p* < 0.001; 18.0% vs. 52.2%, *p* < 0.001; and 5.6% vs. 21.7%, *p* = 0.029, respectively). Worsening of renal function was less frequent in the survivor group than in the non-survivor group (21.3% vs. 60.9%, *p* < 0.001).

Table 4 shows the results of the univariate logistic regression analysis and the multivariate stepwise regression analysis for predicting in-hospital mortality. Only variables that had significant differences (*p*-value < 0.1), except for age and male sex, are shown in the results of the univariate logistic regression analysis (Appendix A). The multivariate stepwise regression analysis model showed that a poor SOB-ASAP score (per point increase; odds ratio (OR): 1.449, 95% CI: 1.159–1.812, *p* = 0.0010), oral phosphodiesterase 3 inhibitor usage at admission (OR: 14.276, 95% CI: 1.119–182.170, *p* = 0.041), and intravenous antibiotic administration within 48 h of hospitalization (OR: 3.887, 95% CI: 1.142–13.224, *p* = 0.030) were significantly associated with in-hospital mortality.

## 4. Discussion

The three main findings of the present study are as follows: (1) the SOB-ASAP score can predict in-hospital mortality even in very elderly patients with acute heart failure, and (2) in addition to the SOB-ASAP score, use of oral phosphodiesterase 3 inhibitors at admission, and requirement of intravenous antibiotic administration within 48 h of hospitalization were important factors for predicting in-hospital mortality secondary to acute heart failure. 

The SOB-ASAP score, which can predict the clinical outcomes of patients with acute heart failure, was found to be useful and practical. Several previous studies have performed risk assessments for patients with heart failure, including assessments with the Get with the Guideline-Heart Failure (GWTG-HF) risk score that was adopted globally to anticipate the outcomes of acute heart failure [21]. Although the SOB-ASAP score was validated in accordance with previous risk scores such as the GWTG-HF risk score, the SOB-ASAP score includes novel serum parameters such as BNP and N-terminal pro-BNP (NT-pro BNP), which were not considered in the previous studies [14]. Therefore, the SOB-ASAP score may predict the outcomes of hospitalized patients with acute heart failure. Furthermore, one of the greatest advantages of the SOB-ASAP score is that it predicts the clinical outcomes of patients with acute heart failure during the very acute phase of hospitalization [14]. The present study suggests the usefulness of a novel risk scoring system even in very elderly patients with acute heart failure, which could be valuable for physicians treating patients with heart failure.

The present study also underscores the clinical impact of treatment with oral phosphodiesterase 3 inhibitors at admission. Phosphodiesterase 3 inhibitors are often required in patients with heart failure having a severe clinical status [22]. One study showed that despite their hemodynamically beneficial effects, long-term therapy with oral phosphodiesterase 3 inhibitors could increase the morbidity and mortality of patients with severe chronic heart failure [23]. In other words, patients treated with oral phosphodiesterase 3 inhibitors are in a worse clinical situation. Therefore, in-hospital mortality could be frequently observed in patients with acute heart failure requiring oral phosphodiesterase 3 inhibitors.

The association between intravenous antibiotic administration within 48 h of hospitalization and worse clinical outcomes should be discussed further. The present study showed that patients who required intravenous antibiotic administration in the early phase of hospitalization were often suspected of being affected with pneumonia, because patients with worse clinical outcomes had higher serum C-reactive protein levels as well as symptoms similar to that of pneumonia. Furthermore, a previous study showed that the coexistence of comorbidities, such as pneumonia, could increase the risk of mortality in the elderly [24,25]. The requirement of intravenous antibiotic administration in the early phase of hospitalization could lead to acute infections such as pneumonia, and could therefore affect the clinical outcomes of elderly patients with heart failure.

Our study has several limitations. First, because this was a single-center retrospective observational study, there is a risk of selection bias. Second, although both left and right cardiac function may affect the clinical prognosis [26,27], the present study did not comprehensively assess the cardiac function. Moreover, the present study did not show the etiology of heart failure. Furthermore, the prognosis of patients with heart failure having a preserved ejection fraction (HFpEF) was as bad as that of patients with heart failure having a reduced ejection fraction (HFrEF). Additionally, the prevalence of HFpEF was high in the elderly patients with heart failure [28,29]. Therefore, the relationship between LVEF and prognosis should be carefully interpreted. Third, the frailty and nutritional status of patients could affect their prognosis [30,31]; however, we could not obtain sufficient information on the baseline frailty and nutritional status due to the severity of acute heart failure and its emergent clinical setting. Fourth, the severity of acute heart failure itself might affect the selection of treatment. For example, in some cases, physicians may hesitate to administer intravenous diuretics due to unstable hemodynamics. Finally, the present study was observational in nature and had a relatively small study population; therefore, it can be considered as a pilot study whose results need to be confirmed prospectively in further extensive multicenter studies.

In conclusion, a poor SOB-ASAP score, oral phosphodiesterase 3 inhibitor use at admission, and requirement of early intravenous antibiotic administration were significantly associated with in-hospital mortality in elderly patients (aged ≥ 80 years) with acute heart failure. Recognizing patients with severe disease and high SOB-ASAP scores, which is a novel risk scoring system, could help physicians to treat patients with heart failure.

## Figures and Tables

**Figure 1 jcm-10-01468-f001:**
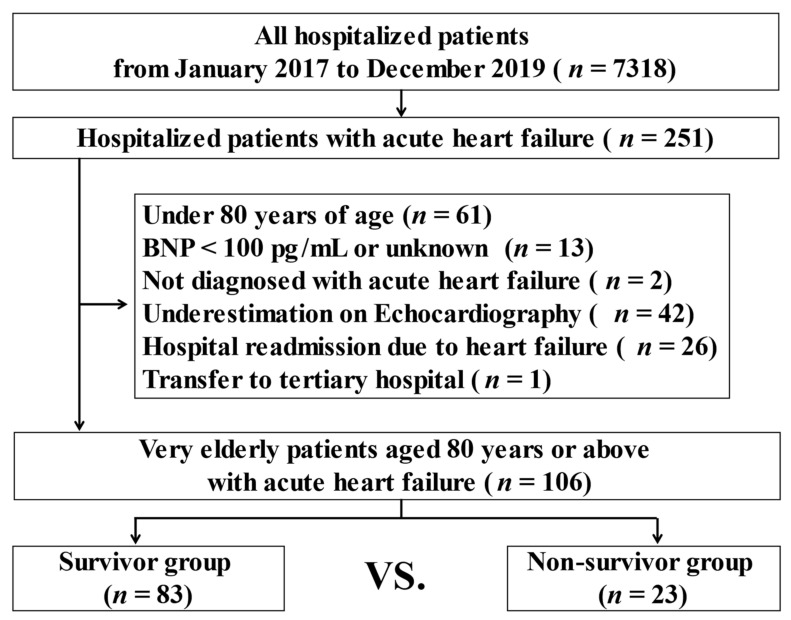
Study flow chart. BNP: brain natriuretic peptide.

**Table 1 jcm-10-01468-t001:** Comparison of the baseline characteristics and medication usage at admission between the survivor and non-survivor groups.

	Survivor Group (*n* = 83)	Non-Survivor Group (*n* = 23)	*p*-Value
Age, years	89.5 ± 4.6	91.0 ± 4.2	0.17
Male sex, *n* (%)	31 (37.3)	8 (34.8)	0.82
Height, cm	147 ± 10	149 ± 9	0.34
Body weight, kg	49.8 ± 10.6	46.4 ± 11.0	0.087
Systolic blood pressure, mmHg (*n*, %)	139 ± 33	119 ± 20	0.011
Diastolic blood pressure, mmHg (*n*, %)	78.5 ± 23.1	72.0 ± 16.0	0.35
Heart rate, beats/minute (*n*, %)	89 ± 26	93 ± 31	0.41
Respiratory rate, breaths/minute (*n*, %)	21 ± 6	19 ± 4	0.13
Systolic blood pressure ≤ 100 mmHg, *n* (%)	7 (8.4)	6 (26.1)	0.022
Hypertension, *n* (%)	76 (91.6)	21 (91.3)	0.62
Dyslipidemia, *n* (%)	21 (25.3)	5 (21.7)	0.48
Diabetes mellitus, *n* (%)	25 (30.1)	6 (26.1)	0.71
Atrial fibrillation/atrial flutter, *n* (%)	46 (55.4)	16 (69.6)	0.22
Pacemaker implantation, *n* (%)	13 (15.7)	1 (4.3)	0.14
Chronic obstructive pulmonary disease, *n* (%)	7 (8.4)	2 (8.7)	0.62
eGFR < 60 mL/min/1.73 m^2^, *n* (%)	61 (73.5)	16 (69.6)	0.71
Ambulance transport to emergency department, *n* (%)	16 (19.3)	6 (26.1)	0.33
NYHA functional classification at admission			0.32
3, *n* (%)	27 (32.5)	5 (21.7)	
4, *n* (%)	56 (67.5)	18 (78.3)	
SOB-ASAP score, (*n*, %)	4.3 ± 2.3	6.8 ± 2.7	<0.001
Medication usage at admission			
ACE-I and/or ARB, *n* (%)	47 (56.6)	9 (39.1)	0.14
β blockers, *n* (%)	27 (32.5)	7 (30.4)	0.85
Calcium channel blockers, *n* (%)	35 (42.2)	7 (30.4)	0.31
Loop diuretics, *n* (%)	47 (56.6)	17 (73.9)	0.13
Mineralocorticoid receptor antagonists, *n* (%)	9 (10.8)	7 (30.4)	0.028
Thiazides, *n* (%)	2 (2.4)	2 (8.7)	0.21
Tolvaptan, *n* (%)	6 (7.2)	3 (13.0)	0.30
Digitalis, *n* (%)	4 (4.8)	1 (4.3)	0.70
PDE3-inhibitor, *n* (%)	1 (1.2)	3 (13.0)	0.031
Statins, *n* (%)	12 (14.5)	3 (13.0)	0.58
Oral anti-diabetes mellitus agents, *n* (%)	12 (14.5)	2 (8.7)	0.37
Anti-platelets, *n* (%)	18 (21.7)	8 (34.8)	0.20
Anti-coagulants, *n* (%)	27 (32.5)	5 (21.7)	0.32

Categorical variables are expressed as numbers and percentages. Continuous variables are expressed as mean ± SD. ACE-I: angiotensin-converting enzyme inhibitor; ARB: angiotensin II receptor blocker; eGFR: estimated glomerular filtration rate; GWTG-HF: Get with the Guideline-Heart Failure; NYHA: New York Heart Association, PDE3: phosphodiesterase 3; SOB-ASAP: SO, serum sodium; B, blood urea nitrogen; A, age and serum albumin; S, systolic blood pressure; and P, natriuretic peptide levels.

**Table 2 jcm-10-01468-t002:** The comparison of baseline laboratory data and echocardiographic parameters between the survivor and non-survivor groups.

	Survivor Group (*n* = 83)	Non-Survivor Group (*n* = 23)	*p*-Value
Laboratory data			
Total protein, g/dL (*n*, %)	6.6 ± 0.7 (80, 96.4)	6.4 ± 0.8 (22, 95.7)	0.22
Serum albumin, g/dL (*n*, %)	3.4 ± 0.5	3.1 ± 0.7	0.10
Total bilirubin, g/dL (*n*, %)	0.79 ± 0.51 (82, 98.8)	0.76 ± 0.40 (22, 95.7)	0.10
Aspartate aminotransferase, U/L (*n*, %)	37 ± 32	83 ± 152	0.70
Alanine aminotransferase, U/L (*n*, %)	23 ± 17	51 ± 93	0.99
Serum sodium, mEq/L (*n*, %)	139 ± 5	136 ± 5	0.025
Serum potassium, mEq/L (*n*, %)	4.2 ± 0.7	4.4 ± 1.0	0.17
Blood glucose, mg/dL (*n*, %)	139 ± 44 (81, 90.4)	151 ± 55 (21, 91.3)	0.12
Blood urea nitrogen, mg/dL (*n*, %)	26.5 ± 16.5	29.1 ± 16.6	0.29
Serum creatinine, mg/dL (*n*, %)	1.24 ± 0.76	1.25 ± 0.60	0.53
Estimated glomerular filtration rate, mL/min/1.73 m^2^ (*n*, %)	47.2 ± 23.3	43.4 ± 20.5	0.59
C-reactive protein, mg/dL (*n*, %)	1.5 ± 2.3	5.1 ± 7.2	0.0073
Hemoglobin, g/dL (*n*, %)	10.9 ± 2.0	11.0 ± 1.9	0.81
Brain natriuretic peptide, pg/mL (*n*, %)	646.9 ± 586.9	1170.7 ± 1018.8	0.0033
Echocardiography results			
Left ventricular ejection fraction, (%)	52.9 ± 17.5	42.1 ± 19.9	0.018
Interventricular septal thickness, mm	9.7 ± 1.7 (82, 98.8)	9.6 ± 2.9 (22, 95.7)	0.36
Left ventricular end-diastolic diameter, mm	44.9 ± 8.8	45.6 ± 11.6	0.95
Left ventricular end-systolic diameter, mm	32.4 ± 9.4	36.3 ± 12.6	0.30
Posterior left ventricular wall thickness, mm	9.8 ± 1.9 (88, 98.8)	10.0 ± 2.2 (22, 95.7)	0.71
Left ventricular end-diastolic volume, mL	95.9 ± 44.3	103.7 ± 60.9	0.98
Left ventricular end-systolic volume, mL	47.3 ± 32.4	65.6 ± 51.2	0.28
Left atrial diameter, mm	43.1 ± 9.4 (82, 98.8)	41.8 ± 8.2	0.56
Aortic diameter, mm	29.7 ± 4.2 (80, 96.4)	31.8 ± 5.2	0.078
Valvular heart disease, *n* (%)	74 (89.2)	22 (95.7)	0.31
Wall motion abnormality, *n* (%)	9 (10.8)	4 (17.4)	0.30

Categorical variables are expressed as numbers and percentages. Continuous variables are expressed as mean ± standard deviation (SD).

**Table 3 jcm-10-01468-t003:** Comparison of treatment and outcomes during hospitalization between the survivor and non-survivor groups.

	Survivor Group (*n* = 83)	Non-Survivor Group (*n* = 23)	*p*-Value
Details of treatment within 48 h of hospitalization			
Intravenous diuretic administration, *n* (%)	67 (80.7)	13 (56.5)	0.017
Intravenous carperitide administration, *n* (%)	1 (1.2)	0 (0.0)	0.78
Tolvaptan introduction, *n* (%)	5 (6.0)	0 (0.0)	0.29
Intravenous nitric acid administration, *n* (%)	11 (13.0)	1 (4.3)	0.21
Digoxin administration, *n* (%)	3 (3.6)	2 (8.7)	0.30
Intravenous catecholamine support requirement, *n* (%)	1 (1.2)	3 (13.0)	0.031
Oral PDE3-inhibitor and/or catecholamine addition *n* (%)	0 (0.0)	1 (4.3)	0.22
Intravenous antibiotic administration, *n* (%)	12 (14.5)	8 (34.8)	0.033
NPPV support requirement, *n* (%)	4 (4.8)	3 (13.0)	0.17
Morphine use, *n* (%)	2 (2.4)	0 (0.0)	0.61
Details of treatment and results during the entire period of hospitalization			
Intravenous diuretic administration, *n* (%)	74 (83.1)	19 (82.6)	0.58
Intravenous carperitide administration, *n* (%)	1 (1.1)	0 (0.0)	0.80
Tolvaptan introduction, *n* (%)	11 (12.4)	6 (26.1)	0.099
Intravenous nitric acid administration, *n* (%)	13 (14.6)	1 (4.3)	0.17
Digoxin administration, *n* (%)	5 (5.6)	2 (8.7)	0.44
Intravenous catecholamine support requirement, *n* (%)	3 (3.4)	7 (30.4)	<0.001
Oral PDE3-inhibitor and/or catecholamine administration, *n* (%)	3 (3.4)	2 (8.7)	0.27
Intravenous antibiotic administration, *n* (%)	16 (18.0)	12 (52.2)	<0.001
NPPV support requirement, *n* (%)	5 (5.6)	5 (21.7)	0.029
Morphine use, *n* (%)	1 (1.1)	2 (8.7)	0.11
Maximum serum creatinine during hospitalization, mg/dL (*n*, %)	1.47 ± 0.96	2.03 ± 1.10	0.0089
Worsening renal function, *n* (%)	19 (21.3)	14 (60.9)	<0.001

Categorical variables are expressed as numbers and percentages. Continuous variables are expressed as mean ± SD. NPPV: noninvasive positive pressure ventilation; PDE3: phosphodiesterase 3.

**Table 4 jcm-10-01468-t004:** Results of the univariate logistic regression analysis and the multivariate stepwise regression analysis for predicting in-hospital mortality.

Univariate Analysis	OR	95% CI	*p*-Value
Age (per year increase)	1.077	0.969–1.197	0.17
Male sex	0.897	0.340–2.352	0.82
Systolic blood pressure (per mmHg increase)	0.975	0.956–0.993	0.0086
SOB-ASAP score (per point increase)	1.455	1.194–1.774	<0.001
Mineralocorticoid receptor antagonist use at admission	3.597	1.167–11.090	0.026
Phosphodiesterase 3 inhibitor use at admission	12.300	1.214–124.581	0.034
Serum albumin (per g/dL increase)	0.462	0.204–1.044	0.063
Aspartate aminotransferase (per U/L increase)	1.007	0.999–1.015	0.079
Alanine aminotransferase (per U/L increase)	1.012	0.998–1.025	0.083
Serum sodium (per mEq/L increase)	0.917	0.838–1.004	0.060
Serum potassium (per mEq/L increase)	1.716	0.926–3.182	0.086
C-reactive protein (per mg/dL increase)	1.267	1.087–1.486	0.0036
Brain natriuretic peptide (per pg/mL increase)	1.001	1.000–1.001	0.0080
Left ventricular end-systolic volume (per mL increase)	1.012	1.000–1.023	0.046
Left ventricular ejection fraction ≥50%	0.446	0.173–1.147	0.094
Aortic diameter (per mm increase)	1.101	0.997–1.216	0.058
Intravenous diuretic administration within 48 h of hospitalization	0.310	0.116–0.834	0.020
Catecholamine support requirement within 48 h of hospitalization	12.300	1.214–124.581	0.034
Intravenous antibiotic administration within 48 h of hospitalization	3.156	1.100–9.052	0.033
Multivariate Analysis	OR	95% CI	*p*-Value
SOB-ASAP score (per point increase)	1.449	1.159–1.812	0.0010
Phosphodiesterase 3 inhibitor use at admission	14.276	1.119–182.170	0.041
Intravenous antibiotic administration within 48 h of hospitalization	3.887	1.142–13.224	0.030

CI: confidence interval; GWTG-HF: Get with the Guideline-Heart Failure; NPPV: noninvasive positive pressure ventilation; NYHA: New York Heart Association; OR: odds ratio; SOB-ASAP: SO, serum sodium; B, blood urea nitrogen; A, age and serum albumin; S, systolic blood pressure; and P, natriuretic peptide levels.

## Data Availability

The datasets generated and/or analyzed during the current study are not publicly available because the study dataset contains potentially identifying clinical information, but are available from the corresponding author upon reasonable request.

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
