# Peer review of "Determinants of In-Hospital Mortality in Elderly Patients Aged 80 Years or above with Acute Heart Failure: A Retrospective Cohort Study at a Single Rural Hospital"

_jcm, 2021, doi:10.3390/jcm10071468_

Round 1

Reviewer 1 Report

This is a well written article of a well conducted research. However, all results are as expected, at least by me. Of course, it is interesting to control if a prediction rule works well in older patients, in a specific setting. But few readers will be interested in it. The authors point correctly to all the weaknesses of the study. Therefore, it might be that this was only a pilot for a more extensive multicentre protocol. 

Details in a separate file. 

Author Response

Response to the comments from Reviewer #1

Thank you for your careful review of our manuscript. We have attempted to incorporate all of your valuable suggestions. We believe that your suggestions have significantly improved the overall scientific content of this work. Responses to each of your comments are given below:

General comment.

This is a well written article of a well conducted research. However, all results are as expected, at least by me. Of course, it is interesting to control if a prediction rule works well in older patients, in a specific setting. But few readers will be interested in it. The authors point correctly to all the weaknesses of the study. Therefore, it might be that this was only a pilot for a more extensive multicentre protocol.

Response to general comment:

We appreciate your comments and agree with them. We have recognized that because our study was observational and the study population was relatively small, our study might be a pilot for a more elaborate multicenter study. Therefore, we have added the following sentence in the paragraph on the limitations of the study in the Discussion section (please see page 8, lines 232–235).

“Finally, the present study was observational in nature and had a relatively small study population; therefore, it can be considered as a pilot study whose results need to be confirmed prospectively in further extensive multicenter studies.”

Detail comments.

Comment 1.

93 The included variables were found to be significantly different (p<0.05) in univariate.

              Normally, statisticians include in the multivariate all predictors with p<0.1 in the univariate. Because sometimes a predictor ‘emerges’.

Response to comment 1:

We thank the reviewer #1 for their comments and agree with their assessment that variables with p<0.1 in the univariate analysis should be included in the multivariate analysis. Therefore, we have re-analyzed our data for the multivariate analysis and have corrected the relevant text in the manuscript. In line with these changes, we have added the following sentence to the “Statistical analyses” subsection (Methods section, please see page 3, lines 99–101) and have added the results of the analysis to Table 4 (Results section). Moreover, because the study results have changed slightly, we have rewritten parts of the Abstract and Discussion to reflect the new findings (please see the yellow highlighted text in these sections).

“Multivariate stepwise regression analysis was used to evaluate the in-hospital mortality in elderly patients (aged ≥80 years) with acute heart failure; the included variables were found to be significant (p <0.1) using univariate logistic regression analysis. ”

Table 4: The results of the univariate logistic regression analysis and the multivariate stepwise regression analysis for predicting in-hospital mortality

Univariate analysis

OR

95% CI

P-value

Age (per year increase)

1.077

0.969–1.197

0.17

Male sex

0.897

0.340–2.352

0.82

Systolic blood pressure (per mmHg increase)

0.975

0.956–0.993

0.0086

SOB-ASAP score (per point increase)

1.455

1.194–1.774

<0.001

Mineralocorticoid receptor antagonist use at admission

3.597

1.167–11.090

0.026

Phosphodiesterase 3 inhibitor use at admission

12.300

1.214–124.581

0.034

Serum albumin (per g/dL increase)

0.462

0.204–1.044

0.063

Aspartate aminotransferase (per U/L increase)

1.007

0.999–1.015

0.079

Alanine aminotransferase (per U/L increase)

1.012

0.998–1.025

0.083

Serum sodium (per mEq/L increase)

0.917

0.838–1.004

0.060

Serum potassium (per mEq/L increase)

1.716

0.926–3.182

0.086

C-reactive protein (per mg/dL increase)

1.267

1.087–1.486

0.0036

Brain natriuretic peptide (per pg/mL increase)

1.001

1.000–1.001

0.0080

Left ventricular end-systolic volume (per mL increase)

1.012

1.000–1.023

0.046

Left ventricular ejection fraction ≥50%

0.446

0.173– 1.147

0.094

Aortic diameter (per mm increase)

1.101

0.997–1.216

0.058

Intravenous diuretic administration within 48 h of hospitalization

0.310

0.116–0.834

0.020

Catecholamine support requirement within 48 h of hospitalization

12.300

1.214–124.581

0.034

Intravenous antibiotic administration within 48 h of hospitalization

3.156

1.100–9.052

0.033

Multivariate analysis

OR

95 % CI

P-value

SOB-ASAP score (per point increase)

1.449

1.159–1.812

0.0010

Phosphodiesterase 3 inhibitor use at admission

14.276

1.119–182.170

0.041

Intravenous antibiotic administration within 48 h of hospitalization

3.887

1.142–13.224

0.030

CI: confidence interval; GWTG-HF: Get with the Guideline-Heart Failure; NPPV: noninvasive positive pressure ventilation; NYHA: New York Heart Association; OR: odds ratio; SOB-ASAP: SO, serum sodium; B, blood urea nitrogen; A, age and serum albumin; S, systolic blood pressure; and P, natriuretic peptide levels.

Comment 2.

95 Meanwhile, the variables included in the SOB-ASAP scoring system were excluded from the multivariate model, such as systolic blood pressure and BNP.

This is indeed very important! Thanks.

Response to comment 2:

We thank reviewer #1 for their comments.

Comment 3.

I wondered why these elderly people with an EF of 42 would die of heart failure. Therefore, I calculated the mean ejection fraction based on the means of end-systolic and end-diastolic volume. I arrive at 0.36! Of course, we should not calculate a mean based on two other means, I know, but the difference is so enormous, I fear some errors have been made.

This is what the authors write:

Predictor                                                                     Survivors              Non survivors     p

Left ventricular ejection fraction, % (%)                   52.8 ± 17.8               42.1 ± 18.9         0.019

Left ventricular end-diastolic volume, mL                 96.8 ± 44.8               103.7 ± 60.9       0.94

Left ventricular end-systolic volume, mL                 48.2 ± 34.4               65.6 ± 51.2         0.31

This is what I find

96,8

103,7

48,2

65,6

0,502066116

0,367406

Response to comment 3:

We thank reviewer #1 for their careful review of our data. We have rechecked our results and datasets. The left ventricular ejection fraction in our study was calculated by the Teichholz method using the left ventricular end-diastolic volume and the left ventricular end-systolic volume. While we noted a few mistakes in our analysis, most of the numerical data remained unchanged. We speculate that the difference shown by reviewer #1 may have been derived from statistic reasons, because our data could not always be normally distributed. Therefore, we have reviewed and shown the results in Table 2 (please see Table 2) and have added the following sentence to the Results section (please see page 5, lines 141–142.).

“The LVEF value was significantly better in the survivor group than in the non-survivor group (52.9 ± 17.5 % vs. 42.1 ± 19.9 %, p = 0.018).”

Table 2: The comparison of baseline laboratory data and echocardiographic parameters between the survivor and non-survivor groups

Survivor group

(n = 83)

Non-survivor group

(n = 23)

P-value

Echocardiography results

Left ventricular ejection fraction, (%)

52.9 ± 17.5

42.1 ± 19.9

0.018

Interventricular septal thickness, mm

9.7 ± 1.7 (82, 98.8)

9.6 ± 2.9 (22, 95.7)

0.36

Left ventricular end-diastolic diameter, mm

44.9 ± 8.8

45.6 ± 11.6

0.95

Left ventricular end-systolic diameter, mm

32.4 ± 9.4

36.3 ± 12.6

0.30

Posterior left ventricular wall thickness, mm

9.8 ± 1.9 (88, 98.8)

10.0 ± 2.2 (22, 95.7)

0.71

Left ventricular end-diastolic volume, mL

95.9 ± 44.3

103.7 ± 60.9

0.98

Left ventricular end-systolic volume, mL

47.3 ± 32.4

65.6 ± 51.2

0.28

Furthermore, we have re-analyzed our data for the univariate and multivariate analyses. Reviewer #3 suggested that we should divide the LVEF into two or three discrete ranges. Therefore, we analyzed the relationship between LVEF and in-hospital mortality by constructing a receiver operating characteristics curve and calculating the area under the curve. The area under the curve was 0.34 (95% confidence interval: 0.21–0.47, P =0.018). An LVEF of 50% had a sensitivity and specificity of 39.1% and 60.2%, respectively. Therefore, we have divided the LVEF into two ranges according to a cut-off LVEF value of 50%. Multivariate analysis revealed that an LVEF ≥ 50% was not significantly associated with in-hospital mortality in elderly patients with acute heart failure. Therefore, we have added the following sentence to the “Statistical analyses” subsection (please see page 3, lines 102–106) and have shown the results of the analysis in Table 4 (please see Table 4).

“We analyzed the relationship between LVEF and the in-hospital mortality by constructing a receiver operating characteristics (ROC) curve and calculating the area under the curve. The area under the curve was 0.34 (95% confidence interval [CI]: 0.21–0.47, P =0.018). The sensitivity and specificity for an LVEF of 50% were 39.1% and 60.2%, respectively. Therefore, an LVEF ≥50% was analyzed in the univariate analysis.”

Table 4: The results of the univariate logistic regression analysis and the multivariate stepwise regression analysis for predicting in-hospital mortality

Univariate analysis

OR

95% CI

P-value

Age (per year increase)

1.077

0.969–1.197

0.17

Male sex

0.897

0.340–2.352

0.82

Systolic blood pressure (per mmHg increase)

0.975

0.956–0.993

0.0086

SOB-ASAP score (per point increase)

1.455

1.194–1.774

<0.001

Mineralocorticoid receptor antagonist use at admission

3.597

1.167–11.090

0.026

Phosphodiesterase 3 inhibitor use at admission

12.300

1.214–124.581

0.034

Serum albumin (per g/dL increase)

0.462

0.204–1.044

0.063

Aspartate aminotransferase (per U/L increase)

1.007

0.999–1.015

0.079

Alanine aminotransferase (per U/L increase)

1.012

0.998–1.025

0.083

Serum sodium (per mEq/L increase)

0.917

0.838–1.004

0.060

Serum potassium (per mEq/L increase)

1.716

0.926–3.182

0.086

C-reactive protein (per mg/dL increase)

1.267

1.087–1.486

0.0036

Brain natriuretic peptide (per pg/mL increase)

1.001

1.000–1.001

0.0080

Left ventricular end-systolic volume (per mL increase)

1.012

1.000–1.023

0.046

Left ventricular ejection fraction ≥50%

0.446

0.173– 1.147

0.094

Aortic diameter (per mm increase)

1.101

0.997–1.216

0.058

Intravenous diuretic administration within 48 h of hospitalization

0.310

0.116–0.834

0.020

Catecholamine support requirement within 48 h of hospitalization

12.300

1.214–124.581

0.034

Intravenous antibiotic administration within 48 h of hospitalization

3.156

1.100–9.052

0.033

Multivariate analysis

OR

95 % CI

P-value

SOB-ASAP score (per point increase)

1.449

1.159–1.812

0.0010

Phosphodiesterase 3 inhibitor use at admission

14.276

1.119–182.170

0.041

Intravenous antibiotic administration within 48 h of hospitalization

3.887

1.142–13.224

0.030

CI: confidence interval; GWTG-HF: Get with the Guideline-Heart Failure; NPPV: noninvasive positive pressure ventilation; NYHA: New York Heart Association; OR: odds ratio; SOB-ASAP: SO, serum sodium; B, blood urea nitrogen; A, age and serum albumin; S, systolic blood pressure; and P, natriuretic peptide levels.

Reviewer 2 Report

The study deals with an important topic in heart disease, which is heart failure. A risk scales allow for more effective standarization of treatment options for patients.

Author Response

Response to the comments from Reviewer #2

Thank you for your careful review of our manuscript. We have attempted to incorporate all of your valuable suggestions. We believe that your suggestions have significantly improved the overall scientific content of this work. Responses to each of your comments are given below:

Comment 1.

The study deals with an important topic in heart disease, which is heart failure. A risk scales allow for more effective standarization of treatment options for patients.

Response to comment 1:

We appreciate reviewer #2’s careful review and their insightful comments. However, the study population of the present study was so small that we could not generate a risk scale for treatment options for heart failure patients. Moreover, we have recognized that our study might be a pilot for further multicenter protocol studies. Therefore, we have added the following sentence to the paragraph on the limitations of the study in the Discussion section (please see page 8, lines 232–235).

“Finally, the present study was observational in nature and had a relatively small study population; therefore, it can be considered as a pilot study whose results need to be confirmed prospectively in further extensive multicenter studies.”

Reviewer 3 Report

I read with interest your paper entitled "Determinants of in-hospital mortality in elderly patients aged 80 years or older with acute heart failure: A retrospective cohort study at a single rural hospital". The novelty, with respect to previous papers, is the use of a simple score named SOB-ASAP to predict outcome of elderly patients admitted at the hospital with acute heart failure. The tool is simple, and even far from offering ultimate results since the retrospective nature of the research and the small sample, and can be easily applied in the daily clinical context. I have 1 question for the authors: how do you explain that patients with a worse outcome were less treated with iv diuretics  (table 2)? Please explain. 

Moreover, when performing logistic regression analysis on continuous variables, such as EF, it is useful to use di or tricothomic split. I suggest to split EF in 2 or 3 discrete ranges (i.e <40 and >40%; or using a cut-off from ROCC analysis.) I strongly suggest the authors to modify the manuscript after these corrections in order to more easily follow the signficance of EF in the evaluation.

Author Response

Response to the comments from Reviewer #3

Thank you for your careful review of our manuscript. We have attempted to incorporate all of your valuable suggestions. We believe that your suggestions have significantly improved the overall scientific content of this work. Responses to each of your comments are given below:

Comment 1.

I read with interest your paper entitled "Determinants of in-hospital mortality in elderly patients aged 80 years or older with acute heart failure: A retrospective cohort study at a single rural hospital". The novelty, with respect to previous papers, is the use of a simple score named SOB-ASAP to predict outcome of elderly patients admitted at the hospital with acute heart failure. The tool is simple, and even far from offering ultimate results since the retrospective nature of the research and the small sample, and can be easily applied in the daily clinical context. I have 1 question for the authors: how do you explain that patients with a worse outcome were less treated with iv diuretics (table 2)? Please explain. 

Response to comment 1:

We thank reviewer #3 for their careful review of our manuscript. We found that patients with a worse outcome were less frequently treated with intravenous diuretics. We believe that this result could reflect the relatively severe clinical situation of patients with acute heart failure. In fact, unstable hemodynamic conditions (such as hypotension) were frequently observed in patients with a worse outcome in the present study. The prevalence of systolic blood pressure ≤ 100 mmHg was higher in the non-survivor group than in the survivor group (26.1% vs. 8.4%, p = 0.022). Therefore, for improved clarity, we have added the following sentences to the Results (please see page 3, lines 121– page 4, line 123) and Discussion (please see page 8, lines 230–232) sections and the results to Table 1 (please see Table 1).

“A systolic blood pressure of ≤ 100 mmHg, indicating an unstable hemodynamic status, was more prevalent in the non-survivor group as compared to in the survivor group (26.1% vs. 8.4%, p = 0.022).”

“Fourth, the severity of acute heart failure itself might affect the selection of treatment. For example, in some cases, physicians may hesitate to administer intravenous diuretics due to unstable hemodynamics.”

Table 1: Comparison of the baseline characteristics and medication usage at admission between the survivor and non-survivor groups

Survivor group

(n = 83)

Non-survivor group

(n = 23)

P-value

Systolic blood pressure ≤ 100 mmHg, n (%)

7 (8.4)

6 (26.1)

0.022

Comment 2.

Moreover, when performing logistic regression analysis on continuous variables, such as EF, it is useful to use di or tricothomic split. I suggest to split EF in 2 or 3 discrete ranges (i.e <40 and >40%; or using a cut-off from ROC analysis.) I strongly suggest the authors to modify the manuscript after these corrections in order to more easily follow the signficance of EF in the evaluation.

Response to comment 2:

We agree with reviewer #3’s assessment that the left ventricular ejection fraction (LVEF) value should be split into two or three discrete ranges according to the cut-off from receiver operating characteristics analysis, in order to present the significance of ejection fraction in the evaluation more clearly. Therefore, we analyzed the relationship between LVEF and in-hospital mortality by constructing an ROC curve and calculating the area under the curve. The area under the curve was 0.34 (95% confidence interval: 0.21–0.47, P =0.018). An LVEF of 50% had a sensitivity and specificity of 39.1% and 60.2%, respectively. Therefore, we used an LVEF ≥ 50% in the univariate and multivariate analyses instead of the degree of LVEF. In conclusion, an LVEF ≥ 50% was not associated with in-hospital mortality in elderly patients with acute heart failure. Accordingly, we have added the following sentence to the “Statistical analyses” subsection (please the Methods section, page 3, lines 102–106) and have presented the results in Table 4.

“We analyzed the relationship between LVEF and the in-hospital mortality by constructing a receiver operating characteristics (ROC) curve and calculating the area under the curve. The area under the curve was 0.34 (95% confidence interval [CI]: 0.21–0.47, P =0.018). The sensitivity and specificity for an LVEF of 50% were 39.1% and 60.2%, respectively. Therefore, an LVEF ≥50% was analyzed in the univariate analysis.”

Table 4: The results of the univariate logistic regression analysis and the multivariate stepwise regression analysis for predicting in-hospital mortality

Univariate analysis

OR

95% CI

P-value

Age (per year increase)

1.077

0.969–1.197

0.17

Male sex

0.897

0.340–2.352

0.82

Systolic blood pressure (per mmHg increase)

0.975

0.956–0.993

0.0086

SOB-ASAP score (per point increase)

1.455

1.194–1.774

<0.001

Mineralocorticoid receptor antagonist use at admission

3.597

1.167–11.090

0.026

Phosphodiesterase 3 inhibitor use at admission

12.300

1.214–124.581

0.034

Serum albumin (per g/dL increase)

0.462

0.204–1.044

0.063

Aspartate aminotransferase (per U/L increase)

1.007

0.999–1.015

0.079

Alanine aminotransferase (per U/L increase)

1.012

0.998–1.025

0.083

Serum sodium (per mEq/L increase)

0.917

0.838–1.004

0.060

Serum potassium (per mEq/L increase)

1.716

0.926–3.182

0.086

C-reactive protein (per mg/dL increase)

1.267

1.087–1.486

0.0036

Brain natriuretic peptide (per pg/mL increase)

1.001

1.000–1.001

0.0080

Left ventricular end-systolic volume (per mL increase)

1.012

1.000–1.023

0.046

Left ventricular ejection fraction ≥50%

0.446

0.173– 1.147

0.094

Aortic diameter (per mm increase)

1.101

0.997–1.216

0.058

Intravenous diuretic administration within 48 h of hospitalization

0.310

0.116–0.834

0.020

Catecholamine support requirement within 48 h of hospitalization

12.300

1.214–124.581

0.034

Intravenous antibiotic administration within 48 h of hospitalization

3.156

1.100–9.052

0.033

Multivariate analysis

OR

95 % CI

P-value

SOB-ASAP score (per point increase)

1.449

1.159–1.812

0.0010

Phosphodiesterase 3 inhibitor use at admission

14.276

1.119–182.170

0.041

Intravenous antibiotic administration within 48 h of hospitalization

3.887

1.142–13.224

0.030

CI: confidence interval; GWTG-HF: Get with the Guideline-Heart Failure; NPPV: noninvasive positive pressure ventilation; NYHA: New York Heart Association; OR: odds ratio; SOB-ASAP: SO, serum sodium; B, blood urea nitrogen; A, age and serum albumin; S, systolic blood pressure; and P, natriuretic peptide levels.

Round 2

Reviewer 1 Report

Thanks for revision.

I still have problems with the ejection fraction. For women, the lower limit of normality is 53%. How can you have a mean of 52% in your surviving group? Did they really have heart failure, or rather pulmonary infection, what the new predictor, antibiotic use, suggests?

And still, I do not understand why patients with EF>40 should die of heart failure …

I have even more problems with the ROC, which the third reviewer proposed. You find an AUC <50%! Please turn your ROC in the other direction. Moreover, if we revert it, 66% is very low! I wonder also how you arrived at the cut-of point of 50%!

Finally, I’d like to see the list of the individual EF of all patients, with their final outcome. And I’d like to see the (correct) ROC.

Author Response

Response to the comments from Reviewer #1

Thank you for carefully reviewing our manuscript. We have incorporated all your valuable suggestions. We believe that your suggestions have significantly improved the overall scientific content of this work. Point-by-point responses to your comments are provided below:

Comment 1:

I still have problems with the ejection fraction. For women, the lower limit of normality is 53%. How can you have a mean of 52% in your surviving group? Did they really have heart failure, or rather pulmonary infection, what the new predictor, antibiotic use, suggests? And still, I do not understand why patients with EF>40 should die of heart failure.

Response to comment 1:

We appreciate your comments and questions. We recognize that because our study had an observational nature, it was liable to selection bias. Therefore, we selected patients with BNP levels >100 pg/mL and with symptoms consistent with heart failure. Furthermore, acute infections, such as pneumonia, exacerbated the symptoms of heart failure. Our study suggested that patients with severe acute heart failure required antibiotics for the concurrent infection. A previous study showed that the prognosis of patients with heart failure having a preserved ejection fraction (HFpEF, EF ≥ 50%) was as bad as that of patients with heart failure having a reduced ejection fraction (HFrEF, EF < 40%). In addition, the prevalence of HFpEF was high in the elderly patients with heart failure. Our results were consistent with those of previous studies. Therefore, we have added the following sentence in the paragraph on the study limitations in the Discussion section (please see page 8,lines 224–227).

“Furthermore, the prognosis of patients with heart failure having a preserved ejection fraction (HFpEF) was as bad as that of patients with heart failure having a reduced ejection fraction (HFrEF). In addition, the prevalence of HFpEF was high in the elderly patients with heart failure [28, 29].”

Comment 2:

I have even more problems with the ROC, which the third reviewer proposed. You find an AUC <50%! Please turn your ROC in the other direction. Moreover, if we revert it, 66% is very low! I wonder also how you arrived at the cut-of point of 50%! Finally, I’d like to see the list of the individual EF of all patients, with their final outcome. And I’d like to see the (correct) ROC.

Response to comment 2:

We appreciate your comments.  Accordingly, we have calculated our ROC in the correct (reverse) direction. The ROCs have been revised and shown below. Moreover, reviewer #1 suggested that the AUC was not high, and this may have been affected by the study’s small sample size. Therefore, we have added the following sentence in the paragraph of the statistical analyses section and study limitations in the Discussion section (please see page 3, lines 101–105 and page 8, lines 234–236). Furthermore, we have provided details of LVEF, LVEDV, LVESV, and in-hospital mortality in response to reviewer #1’s request (Please see the table below).

“We analyzed the relationship between LVEF and the in-hospital mortality by constructing a receiver operating characteristics (ROC) curve and calculating the area under the curve. The area under the curve was 0.66 (95% confidence interval [CI]: 0.53–0.79, P =0.018). The sensitivity and specificity for an LVEF of 49.6% were 60.2% and 39.1%, respectively. Therefore, an LVEF ≥50% was analyzed in the univariate analysis.”

“Finally, the present study was observational in nature and had a relatively small study population; therefore, it can be considered as a pilot study whose results need to be confirmed prospectively in further extensive multicenter studies.”

ROC curve. The red arrow shows the LVEF point of 49.6% (sensitivity 60.2%, specificity 39.1%).

LVEF, LVEDV, LVESV and in-hospital mortality in the overall study population

study number

In-hospital mortality_1

LVEF_%

LVEDV_ml

LVESV_ml

1

0

26

177

131

2

0

76.5

54.1

12.7

3

0

28.3

198

142

4

0

30.8

91

63

5

0

44.8

91

50.2

6

1

23.1

221

170

7

0

45.5

113

61.6

8

0

72.1

50

14

9

0

64.4

57.8

20.6

10

0

33.9

66.3

43.8

11

0

54.9

37.9

17.1

12

0

32.5

110

74.2

13

1

41.1

53.3

31.4

14

0

57.5

80.4

34.2

15

0

86.4

74.2

10.1

16

0

55.2

70.8

31.7

17

1

42.9

198

113

18

1

44.3

74.7

41.6

19

0

59.5

95.9

38.8

20

0

56.5

88.6

38.5

21

0

55.2

70.8

31.7

22

0

34.2

199

131

23

0

67.6

92

29.8

24

0

66.3

134

45.1

25

1

26.7

202

148

26

0

20.3

119

94.9

27

1

15.1

47.1

40

28

0

55.2

88.6

39.7

29

1

20.5

185

147

30

0

52.5

65.1

30.9

31

0

78.8

91.4

19.4

32

0

41.9

78.1

45.4

33

0

64.8

130

45.8

34

1

69.4

96.8

29.6

35

1

23.1

83.1

63.9

36

0

40.5

80.4

47.8

37

0

66.2

72.1

24.4

38

0

43.1

165

93.9

39

1

46.3

84

45.1

40

0

61.7

92.9

35.6

41

0

67.4

32.5

10.6

42

0

63.3

97.8

35.9

43

0

60

92.4

37

44

0

55.6

110

48.8

45

0

70.6

131

38.5

46

0

23.2

98.9

75.9

47

0

67.5

77.3

25.1

48

0

50.4

46.4

23

49

0

57.8

145

61.2

50

0

42.3

134

77.3

51

0

73.8

32.5

8.5

52

0

56

97.3

42.8

53

1

59.9

49.1

19.7

54

0

71.2

60.8

17.5

55

1

58.1

161

67.5

56

0

64.1

230

82.6

57

0

31.4

242

166

58

1

71

35.9

10.4

59

0

24.8

105

79

60

0

62.1

128

48.5

61

0

61

110.7

43.2

62

0

41.3

165

96.8

63

0

68.2

33.3

10.6

64

1

65.9

83.1

28.3

65

0

54.1

33.3

15.3

66

0

60.3

55.2

21.9

67

1

23.7

198

151

68

0

30.4

111

77.3

69

0

71.2

68.3

19.7

70

0

77.7

44.8

10

71

0

77.1

74.2

17

72

0

39.7

119

71.7

73

0

49.8

93.9

47.1

74

0

67.6

89.6

29

75

0

23.2

70.4

54.1

76

0

66.7

92.8

30.9

77

1

20.6

117

92.9

78

0

49.3

126

63.9

79

0

17.2

37.9

31.4

80

0

27.4

82.6

60

81

0

60.9

72.9

28.5

82

0

64.8

65.9

23.2

83

0

71.3

58.5

16.8

84

0

45.8

140

75.9

85

0

30

39.7

27.8

86

1

14.8

135

115

87

1

40.2

58.5

35

88

0

77.6

28

6.27

89

0

77.2

88.6

20.2

90

1

16

100

84

91

0

28.1

134

96.3

92

0

55.8

81.3

35.9

93

0

30.3

118

82.2

94

0

20.6

114

90.5

95

0

37.5

110

68.8

96

1

53.4

40.6

18.9

97

0

47

31.5

16.7

98

1

68.9

74

23

99

0

43.1

62

35.3

100

0

76.4

118

27.8

101

1

61

43.8

17.1

102

1

61.4

44.8

17.3

103

0

36.3

131

83.5

104

0

41

64.7

38.2

105

0

76.2

146

34.7

106

0

71

150

43.5

Reviewer 3 Report

I appreciated the corrections you made.

Best regards FP

Author Response

Response to the comments from Reviewer #3

Thank you for carefully reviewing our manuscript.

Comment 1.

I appreciated the corrections you made. Best regards FP.

Response to comment 1:

Thank you very much. We are grateful for your comments that helped us to greatly improve our manuscript.